# SoftAdam: Unifying SGD and Adam for better stochastic gradient descent

## Abstract

Stochastic gradient descent (SGD) and Adam are commonly used to optimize deep neural networks, but choosing one usually means making tradeoffs between speed, accuracy and stability. Here we present an intuition for why the tradeoffs exist as well as a method for unifying the two in a continuous way. This makes it possible to control the way models are trained in much greater detail. We show that for default parameters, the new algorithm equals or outperforms SGD and Adam across a range of models for image classification tasks and outperforms SGD for language modeling tasks.

## 1 Introduction

One of the most common methods of training neural networks is stochastic gradient descent (SGD) (Bottou et al. (2016)). SGD has strong theoretical guarantees, including convergence in locally non-convex optimization problems (Lee et al. (2016)). It also shows improved generalization and stability when compared to other optimization algorithms (Smith & Le (2018)).

There have been various efforts in improving the speed and generalization of SGD. One popular modification is to use an adaptive gradient (Duchi et al. (2011)), which scales the gradient step size to be larger in directions with consistently small gradients. Adam, an implementation that combines SGD with momentum and an adaptive step size inversely proportional to the RMS gradient, has been particularly successful at speeding up training and solving particular problems (Kingma & Ba (2014)). However, at other problems it pays a penalty in worse generalization (Wilson et al. (2017); Keskar & Socher (2017)), and it requires additional modifications to achieve a convergence guarantee (Reddi et al. (2018); Li & Orabona (2018)).

Here we develop an intuition for adaptive gradient methods that allows us to unify Adam with SGD in a natural way. The new optimizer, SoftAdam, descends in a direction that mixes the SGD with Adam update steps. As such, it should be able to achieve equal or better optimization results across a variety of problems.

### 1.1 Related Work

Several authors have recently tried to combine Adam and SGD to get the best of both worlds. However, these have not always enabled better generalization or performance across different problems.

In one study, the optimization algorithm was switched from Adam to SGD during training based on a scale-free criterion, preventing the addition of a new hyper-parameter (Keskar & Socher (2017)). The result is that the longer the convolutional networks were trained on Adam, the worse their generalization performance compared to SGD. The best performance came from switching to SGD in this case as soon as possible.

Another recent algorithm takes the approach of clipping the large Adam updates to make them more similar to SGD as the training nears the end (Luo et al. (2019)). However, this approach requires two new hyper-parameters: the rate at which the training is switched over, and a learning rate for both SGD and Adam.

Similar to this work, partially adaptive methods (Chen & Gu (2018)) can allow arbitrary mixing between SGD and Adam. However, in that work the step size is not strictly smaller than the SGD

step and so the same guarantees cannot be made about convergence. It is of interest to see whether there is any advantage over these methods.

Other algorithms have shown improvements on SGD by averaging weights over many steps (Polyak & Juditsky (1992); Zhang et al. (2019); Izmailov et al. (2018)). These algorithms are complementary to the algorithm developed here, as they require an underlying algorithm to choose the step direction at any point.

## 2 SGD AND ADAM

The fundamental idea of gradient descent is to follow the path of steepest descent to an optimum. Stochastic gradient descent enables us to optimize much larger problems by using randomly sub-sampled training data. The stochastic gradient descent algorithm will minimize the loss function $J(\theta; x)$, which is parameterized by $\theta$ and takes as input training data $x$,

$$g_t \leftarrow [\nabla_\theta J(\theta; x_t)]_{\theta=\theta_{t-1}}$$
$$\theta_t \leftarrow \theta_{t-1} - \alpha g_t,$$

where $\alpha$ is a learning rate that may vary with $t$ and $x_t$ is the training data selected for the batch at step $t$. The convergence rate can be improved further by using a running average of the gradient, initializing with $m_0 \leftarrow 0$. This method, known as momentum (Goh (2017)), may be written as,

$$g_t \leftarrow [\nabla_\theta J(\theta; x_t)]_{\theta=\theta_{t-1}}$$
$$m_t \leftarrow \beta_1 m_{t-1} + (1 - \beta_1) g_t,$$
$$\theta_t \leftarrow \theta_{t-1} - \alpha m_t,$$

A further development that has improved convergence, especially for LSTM and language modeling tasks, involves the second gradient as well. This specific version is known as the Adam algorithm (Kingma & Ba (2014)),

$$g_t \leftarrow [\nabla_\theta J(\theta; x_t)]_{\theta=\theta_{t-1}}$$
$$m_t \leftarrow \beta_1 m_{t-1} + (1 - \beta_1) g_t,$$
$$v_t \leftarrow \beta_2 v_{t-1} + (1 - \beta_2) g_t^2,$$
$$\theta_t \leftarrow \theta_{t-1} - \frac{\alpha \hat{m}_t}{\sqrt{\hat{v}_t}},$$

where $\hat{m}_t = m_t / \sqrt{1 - \beta_1^t}$ and $\hat{v}_t = v_t / \sqrt{1 - \beta_2^t}$ are unbiased estimators of the first and second moment respectively.

### 2.1 CONVERGENCE

In order to analyze the convergence of these algorithms, we can consider a second-order approximation of $J$ on its combined argument $z = (\theta; x)$ in the region of $(\theta_t; x_t)$,

$$J(z) = \frac{1}{2} z^T H_t z - b^T z,$$

where $H_t$ is the Hessian of $J(z)$ around $z_t$. This gives us the gradient,

$$g_t = \nabla J(z_t) = H_t z_t - b,$$

which becomes the SGD update step,

$$z_{t+1} = z_t - \alpha (H_t z - b).$$

Unrolling this update step can be shown to lead to an expression for the distance from the optimal value $z^\star$, in the basis of Hessian eigenvectors $\xi_i$:

$$z_t - z^\star = \sum_i (z_0 \cdot \xi_i)(1 - \alpha\lambda_i)^t \xi_i.$$

We can see that the learning is stable if the learning rate $\alpha$ satisfies,

$$|1 - \alpha\lambda_i| < 1.$$

In addition, we find that the value for the learning rate that leads to the fastest overall convergence is,

$$\alpha = \frac{2}{\lambda_1 + \lambda_n}, \tag{1}$$

where $\lambda_1$ and $\lambda_n$ are the max and min eigenvalues of $H$, respectively.

## 2.2 ADAPTIVE MOMENT ESTIMATION

If rather than a single learning rate $\alpha$, we were to use a diagonal matrix $D$ such that the update is,

$$z_{t+1} = z_t - Dg_t,$$

we may be able to modify the diagonal entries $[d_0, d_1, \ldots, d_n]$ such that faster overall convergence is achieved. For example, in the special case that the Hessian is diagonal, the convergence rate for the $i$-th element becomes,

$$r_i = (1 - d_i\lambda_i),$$

which has the solution $d_i = \lambda_i^{-1}$. In this situation, if the eigenvalues $\lambda_i$ are known, the algorithm can converge to the minimum in exactly one step. This corresponds with some intuition behind adaptive moment methods: that taking a step with a "constant" size in every direction toward the target will reach convergence faster than taking a step proportional to the gradient size.

Because the eigenvalues and eigenvectors not known *a priori*, for a practical algorithm we must rely on an approximation to find $d_i$. One technique named AdaGrad (Duchi et al. (2011)) prescribes the diagonal elements:

$$d_i = \frac{\alpha}{\epsilon + \sqrt{\sum_t g_{ti}^2}}. \tag{2}$$

For building our intuition, we consider the special case where the Hessian is diagonal,

$$g_{ti}^2 = (\lambda_i z_{ti} - b_i)^2.$$

Combining this with Equation 2, we compare the AdaGrad coefficient to the optimal value for $d_i = \lambda_i^{-1}$, finding,

$$\frac{\alpha}{\epsilon + \sqrt{\sum_t (\lambda_i z_{ti} - b_i)^2}} \sim \frac{1}{\lambda_i}. \tag{3}$$

As long as $\epsilon \ll \alpha\lambda_i$, this will be true when,

$$\alpha^2 \sim \sum_t \left(z_{ti} - \frac{b_i}{\lambda_i}\right)^2 \sim \text{constant w.r.t.}\, i, \tag{4}$$

which is the sum squared distance of the parameter value $z_{ti}$ to its optimum $z_i^\star = b_i/\lambda_i$. This reveals some of the tradeoffs being made by AdaGrad. It will perform ideally when all parameters start the same distance from their targets.

This may be true on average if the initializations $z_{0i}$ and optima $z_i^\star$ can be made over the same subspaces. That is, if $z_i^\star$ is uncorrelated to $\lambda_i$, we can expect this to have good performance on average.

However, there can be significant errors in both overestimating and underestimating the eigenvalue. One would expect that for a typical problem $b_i$ and $\lambda_i$ might be drawn from uncorrelated distributions. In this case, large values of $z_i^\star$ will be likely to correspond to small values of $\lambda_i$. Since $z_0$ can only be drawn from the average distribution (no information is known at this point), the estimated $\lambda_i$ is more likely to be large, as the initialization will be far from the optimum. Intuitively, the gradient is large because the optimum is far from the initialization, but the algorithm mistakes this large gradient for a large eigenvalue.

On the other hand, when the parameter is initialized close to its optimum, the algorithm will mistakenly believe the eigenvalue is small, and so take relatively large steps. Although they do not affect the overall convergence much on their own (since the parameter is near its optimum), these steps can add significant noise to the process, making it difficult to accurately measure gradients and therefore find the optimum in other parameters.

This problem will be significantly worse for Adam, which forgets its initialization with some decay factor $\beta_2$. In that case, as each parameter reaches its optimum, its estimated eigenvalue $\lambda_i$ drops and the step size gets correspondingly increased. In fact, the overall algorithm can be divergent as each parameter reaches its optimum, as the step size will grow unbounded unless $\alpha$ is scheduled to decline properly or a bounding method like AMSGrad is used (Reddi et al. (2018)).

In addition, reviewing our earlier assumption of small $\epsilon$, these algorithms will perform worse for small eigenvalues $\lambda_i < \epsilon/\alpha$. This might be especially bad in Adam where late in training when the learning rate $\alpha \sim \sum_t \left( z_{ti} - \frac{b_i}{\lambda_i} \right)^2$ is small.

We finally note that the Hessians in deep learning problems are not diagonal (Sagun et al. (2016); Li et al. (2019)). As such, each element might be better optimized by a learning rate that better serves both its min and max eigenvalues.

## 3    SMOOTHED ADAPTIVE GRADIENT

Overall, this understanding has led us to believe that adaptive moments might effectively estimate $\lambda_i$ when it is large, but might be less effective when it is small. In order to incorporate this information about large eigenvalues, as well as optimize the learning rate to account for variation in the eigenvalues contributing to convergence of a particular component, we consider the an update to Eq. 1,

$$ d_i = \frac{\alpha \bar{\lambda} \left( 1 + \eta \right)}{\bar{\lambda} + \eta \lambda_i}, $$

where $\bar{\lambda}$ is an average eigenvalue and $\eta$ is a new hyper-parameter that controls the weighting of the eigenvalue estimation. Here we have added $\bar{\lambda}$ to the numerator so that $\alpha$ does not need to absorb the changes to the RMS error as it does in Eq. 4. This also recovers the SGD convergence guarantees, since the step is always within a factor of $\eta$ to an SGD step. In addition, this will allow us to recover SGD with momentum exactly in the limit $\eta \to 0$. We use the adaptive gradient estimation,

$$ \frac{\lambda_i^2}{\bar{\lambda}^2} \approx \frac{\langle g_i^2 \rangle}{\epsilon^2 + \langle g^2 \rangle_{avg}} = \frac{v_{ti}}{\epsilon^2 + \bar{v}_t}, $$

where $\bar{v}_t$ is the mean value of $v_t$, to write,

$$ d_i = \frac{\alpha \left( 1 + \eta \right)}{1 + \eta \sqrt{v_{ti} / \left( \epsilon^2 + \bar{v}_t \right)}}. $$

One issue with the above estimation is that its variance is very large at the beginning of training (Liu et al. (2019)). It was suggested that this is the reason that warmup is needed for Adam and shown

---

**Algorithm 1** The SoftAdam Algorithm

---

**Input**: $\theta_0 \in \mathcal{F}$: initial parameters, $\{\alpha_t > 0\}_{t=1}^T$: learning rate, $\alpha_{wd}, \beta_1, \beta_2, \eta, \epsilon$: other hyperparameters, $J_t(\theta)$: loss function
**Output**: $\theta_T \in \mathcal{F}$
$m_0, v_0 \leftarrow 0$                                                             ▷ Initialize moments to 0
**for** $t \leftarrow 1 \ldots T$ **do**
      $g_t \leftarrow \nabla_\theta J(z)$                                                   ▷ Find gradient
      $m_t \leftarrow \beta_1 m_{t-1} + (1 - \beta_1) g_t$                           ▷ Update first moment
      $\hat{\beta}_{2t} \leftarrow \min (\beta_2, 1 - 1/t)$
      $v_t \leftarrow \hat{\beta}_{2t} v_{t-1} + \left(1 - \hat{\beta}_{2t}\right) g_t^2$                  ▷ Update second moment
      $\bar{v}_t \leftarrow \text{mean}_{elem} [v_t]$                                    ▷ Average over all elements
      $r_t \leftarrow \text{sqrt} \left[ (1 - \beta_2) / \left(1 - \hat{\beta}_{2t}\right) \right]$
      $d_t = \alpha (1 + \eta) \left( 1 + \eta - \eta r_t + \eta r_t \sqrt{v_t / (\bar{v}_t + \epsilon^2)} \right)^{-1}$       ▷ Calculate the denominator
      $\theta_t = \theta_{t-1} - m_t d_t$                                             ▷ Perform the update
**end for**
**return** $\theta_T$

---

that rectifying it can make warmup unnecessary. Where $v_t$ is the average of $n_t$ elements and $v_\infty$ the average of $n_\infty$, we define $r_t = \sqrt{n_t/n_\infty}$ and:

$$d_i = \frac{\alpha (1 + \eta)}{1 + \eta - \eta r_t + \eta r_t \sqrt{v_{ti} / (\epsilon^2 + \bar{v}_t)}}.$$

This finally forms the basis for our algorithm.

### 3.1 ALGORITHM

Our algorithm differs from Adam in a few other ways. First, the biased gradient estimate is used rather than the unbiased one. This matches the SGD implementation of momentum, and also avoids magnifying the large variance of early gradient estimates:

$$m_t = \beta_1 m_{t-1} + (1 - \beta_1) g_t.$$

In addition, the second moment $v_t$ is calculated in an unbiased way using an effective $\hat{\beta}_2(t)$, or by abuse of notation $\hat{\beta}_{2t}$:

$$\hat{\beta}_{2t} = \min \left( \beta_2, 1 - \frac{1}{t} \right),$$
$$v_t = \hat{\beta}_{2t} v_{t-1} + \left( 1 - \hat{\beta}_{2t} \right) g_t.$$

This has a negligable impact on the performance of the algorithm, but makes tracking the moment over time easier since it does not need to be un-biased later. We then calculate the ratio of the number of samples $n_t$ used to calculate the moment $v_t$ to the steady state number of samples in the average $n_\infty$:

$$r_t = \sqrt{\frac{n_t}{n_\infty}} = \sqrt{\frac{1 - \beta_2}{1 - \hat{\beta}_{2t}}}.$$

We finally note that the weight decay should be calculated separately from this update as in AdamW (Loshchilov & Hutter (2017)).

Table 1: Top-1 validation accuracy (%) on CIFAR-10 for different AdamW, SGD and SoftAdam. SoftAdam is restricted to $\eta = 1$.

| ARCHITECTURE | ADAMW | SGD | SOFTADAM |
|---|---|---|---|
| AlexNet | $74.55 \pm 0.37$ | $77.10 \pm 0.50$ | $\mathbf{79.30 \pm 0.25}$ |
| VGG19 (BN) | $92.50 \pm 0.25$ | $93.52 \pm 0.18$ | $\mathbf{93.89 \pm 0.12}$ |
| PreResNet-56 | $91.93 \pm 0.18$ | $\mathbf{94.01 \pm 0.05}$ | $\mathbf{94.02 \pm 0.03}$ |
| ResNet-110 | $92.11 \pm 0.04$ | $94.54 \pm 0.29$ | $\mathbf{94.75 \pm 0.15}$ |
| DenseNet-100 | $93.25 \pm 0.19$ | $\mathbf{95.04 \pm 0.08}$ | $\mathbf{95.00 \pm 0.11}$ |

These steps are combined and summarized in Algorithm 1. A reference implementation for the PyTorch package (Paszke et al. (2017)) is provided in Appendix A.

## 4 EMPIRICAL RESULTS

In order to test the ability of this algorithm to reach better optima, we performed testing on a variety of different deep learning problems. In these problems we keep $\eta$ at the default value of 1. Because the algorithm is the same as SGDM if $\eta = 0$ and is comparable to Adam when $\eta = \epsilon^{-1}$, getting results at least as good as those algorithms is just a matter of parameter tuning. These results are intended to show that SoftAdam performs remarkably well with a common parameter choice. For the best performance, the hyper-parameter $\eta$ and learning rate schedule $\alpha$ should be optimized for a particular problem.

### 4.1 CIFAR-10

We trained a variety of networks:[1] AlexNet (Krizhevsky et al. (2012)), VGG19 with batch normalization (Simonyan & Zisserman (2014)), ResNet-110 with bottleneck blocks (He et al. (2015)), PreResNet-56 with bottleneck blocks (He et al. (2016)), DenseNet-BC with L=100 and k=12 (Huang et al. (2016)) on the CIFAR-10 dataset (Krizhevsky (2012)) using SGD, AdamW and SoftAdam. For each model and optimization method, the weight decay was varied over [1e-4,2e-4,5e-4,1e-3,2e-3,5e-3]. For AdamW the learning rate was varied over [1e-4,2e-4,5e-4,1e-3,2e-3]. For each optimizer and architecture and the best result was chosen, and three runs with separate initializations were used go generate the final data. The learning schedule reduced the learning rate by a factor of 10 at 50% and 75% through the total number of epochs. The results are summarized in Table 1. We find that SoftAdam equals or outperforms SGD in training classifiers on this dataset. Due to the larger optimal weight decay constant, SoftAdam achieves lower validation loss at a higher train loss than SGD.

### 4.2 PENN TREEBANK

We trained a 3-layer LSTM with 1150 hidden units per layer on the Penn Treebank dataset (Mikolov et al. (2010)) in the same manner as Merity et al. (2017). For SoftAdam the weight drop was increased from 0.5 to 0.6. Results for the average of three random initializations are shown in Figure 2 (a) and are summarized in Table 2. For these parameters, SoftAdam outperforms SGD significantly but does not quite achieve the same results as Adam. Note that for this experiment we chose Adam instead of AdamW for comparison due to its superior performance.

### 4.3 IWSLT'14 GERMAN TO ENGLISH

We also trained a transformer using the fairseq package by Ott et al. (2019) on the IWSLT'14 German to English dataset. Results for each method with optimized hyperparameters are summarized in Table 3. Note that no warmup is used for training SoftAdam, but warmup is used for AdamW and

---

[1]Implementation was based on the public repository at `https://github.com/bearpaw/pytorch-classification`

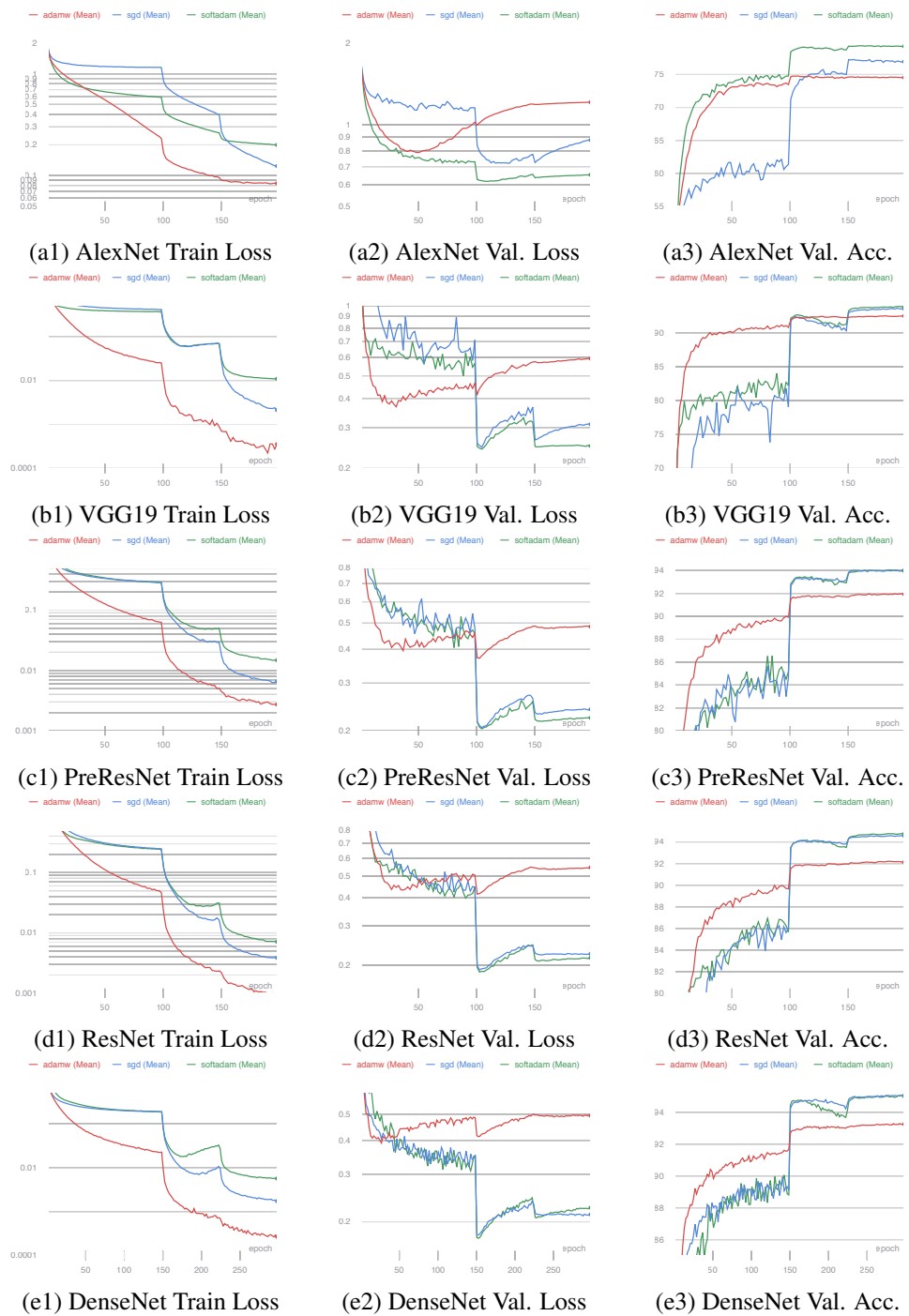

Figure 1: Training results for CIFAR-10. Architectures are (a) AlexNet, (b) VGG19 (BN), (c) PreResNet-56, (d) ResNet-110, (e) DenseNet-100 (BC) using SoftAdam, AdamW and SGD. Each line indicates the average of 3 training runs with random initializations.

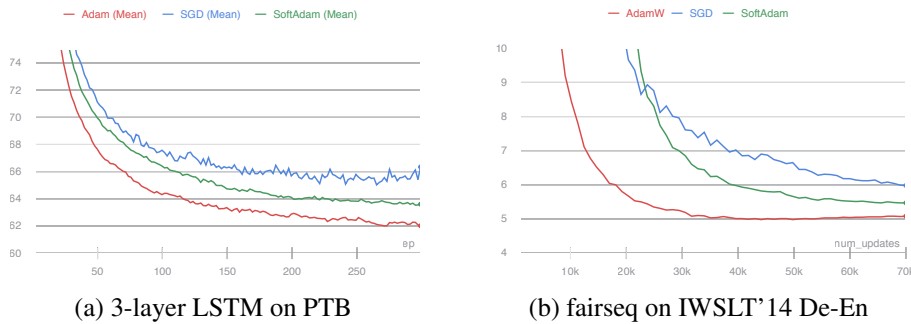

(a) 3-layer LSTM on PTB (b) fairseq on IWSLT'14 De-En

Figure 2: Language model validation perplexity during training for SoftAdam, Adam/AdamW and SGD.

Table 2: LSTM validation perplexity, test perplexity, and test loss on the Penn Treebank dataset for SoftAdam, Adam and SGD.

| OPTIMIZER | VAL. PPL. |
|---|---|
| SoftAdam | $63.63 \pm 0.19$ |
| SGD | $66.64 \pm 0.85$ |
| Adam | $\mathbf{62.03 \pm 0.34}$ |

SGD. Following Liu et al. (2019), we use a linear learning rate schedule. For these parameters, we again find that SoftAdam outperforms SGD significantly.

## 5 CONCLUSIONS

In this paper, we have motivated and demonstrated a new optimization algorithm that naturally unifies SGD and Adam.

We have focused our empirical results on the default hyper-parameter setting, $\eta = 1$, and predetermined learning schedules. With these parameters, the algorithm was shown to produce optimization that is better than or equal to SGD and Adam on image classification tasks. It also performed significantly better than SGD on language modeling tasks.

Together with finding the optimal values for $\eta$, we expect a better understanding of the learning schedule to bring light to the way in which the adaptive gradient methods improve convergence. SoftAdam now also makes it possible to create a learning schedule on $\eta$, which may be another fruitful avenue of research, expanding on the work of Ward et al. (2018).

Better understanding of how adaptive gradients improve the convergence of practical machine learning models during training will enable larger models to be trained to more accurately in less time. This paper provides a useful intuition for how that occurs and provides a new algorithm that can be used to improve performance across a diverse set of problems.

Table 3: Best results for training IWSLT'14 De-En with different optimization algorithms.

| OPTIMIZER | VAL. PPL. | BLEU SCORE |
|---|---|---|
| SGD | 4.14 | 31.43 |
| SoftAdam | 4.02 | 32.76 |
| AdamW | **3.87** | **34.44** |

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

## A  REFERENCE PYTORCH IMPLEMENTATION

```python
import math
import torch
from torch.optim.optimizer import Optimizer
import numpy as np

class SoftAdam(Optimizer):
    def __init__(
        self,
        params,
        lr=1e-1,
        betas=(0.9, 0.999),
        eps=1e-8,
        eta=1.0,
        weight_decay=0,
        nesterov=False,
    ):
        defaults = dict(
            lr=lr,
            betas=betas,
            eps=eps,
            eta=eta,
```

```
                    weight_decay=weight_decay,
                    nesterov=nesterov,
            )
            super(SoftAdam, self).__init__(params, defaults)

    def step(self, closure=None):
        loss = None
        if closure is not None:
            loss = closure()

        for group in self.param_groups:
            for p in group["params"]:
                if p.grad is None:
                    continue
                grad = p.grad.data
                if grad.is_sparse:
                    raise RuntimeError(
                        "SoftAdam does not support sparse gradients"
                    )
                nesterov = group["nesterov"]

                state = self.state[p]

                # State initialization
                if len(state) == 0:
                    state["step"] = 0
                    # Exponential moving average of gradient values
                    state["exp_avg"] = torch.zeros_like(
                        p.data
                    )
                    # Exponential moving average of
                    # squared gradient values
                    state["exp_avg_sq"] = torch.zeros_like(
                        p.data
                    )

                exp_avg, exp_avg_sq = (
                    state["exp_avg"],
                    state["exp_avg_sq"],
                )
                beta1, beta2 = group["betas"]

                state["step"] += 1
                beta2_hat = min(
                    beta2, 1.0 - 1.0 / (state["step"])
                )
                r_beta = (1 - beta2) / (1 - beta2_hat)
                eta_hat2 = (
                    group["eta"] * group["eta"] * r_beta
                )

                # Decay the first and second moment with the
                # running average coefficient
                exp_avg.mul_(beta1).add_(1 - beta1, grad)
                exp_avg_sq.mul_(beta2_hat).addcmul_(
                    1 - beta2_hat, grad, grad
                )

                # Create temporary tensor for the denominator
```

```
denom = exp_avg_sq.mul(
    eta_hat2
    / (
        torch.mean(exp_avg_sq)
        + group["eps"] * group["eps"]
    )
)
denom.sqrt_().add_(
    1 + group["eta"] - np.sqrt(eta_hat2)
)

wd = group["weight_decay"] * group["lr"]

p.data.add_(-wd, p.data)

lr_eff = group["lr"] * (1 + group["eta"])

if nesterov:
    p.data.addcdiv_(
        -lr_eff * beta1, exp_avg, denom
    )
    p.data.addcdiv_(
        -lr_eff * (1 - beta1), grad, denom
    )
else:
    p.data.addcdiv_(-lr_eff, exp_avg, denom)

return loss
```

