# OpenReview forum: "SoftAdam: Unifying SGD and Adam for better stochastic gradient descent"
_ICLR.cc/2020/Conference — Reject_

### Official Review · AnonReviewer1 · 2019-10-20
**Official Blind Review #1**

**Rating:** 3

**Review:**

The paper tends to explain how the tradeoffs between convergence speed and convergence performance are made by different optimization methods. Moreover, the paper modifies Adam-like updating rules and proposes a novel optimization methods, SoftAdam. Finally, the paper performs numerical experiments on traditional image classification tasks as well as language modeling tasks.

Pros
The paper involves the language modeling tasks in empirical results besides traditional image classification tasks, which helps to explain the convergence performance of optimization methods in a wider range of applications.

Cons
1. The writing of this paper is not well organised. Section 1 lacks detailed description of the main idea and the proposed optimization methods, which actually confuses the reader. Section 2 describes too much details of SGD and Adam, and lacks a clear "intuition" which readers exactly expect.
2. The notation in the paper is little confusing. In the update rule of z_{t+1} in Section 2, what is meaning of z? In Section 3, n_t and n_\infty are used before a proper defination, and what is relationship between \alpha and \alpha_t in the implementation of the SoftAdam?
3. The motivation of the proposed method is weak. Such a weak motivation is mainly because of the insufficient "intuition" in Section 2. The author mentions "the Hessians in deep learning problems are not diagonal", but does not provide further explanation on why more importance should be lay on serving both max and min eigenvalues.
4. There are also several minor problems on the numerical results. Firstly, why the colors of "softAdam" and "sgd" are switched several times in Figure 1? Secondly, the figural result in Figure 1 and he numerical results on language processing models both lack a display of the confidence range.


**Experience Assessment:**

I have published in this field for several years.

**Review Assessment: Checking Correctness Of Derivations And Theory:**

I assessed the sensibility of the derivations and theory.

**Review Assessment: Checking Correctness Of Experiments:**

I assessed the sensibility of the experiments.

**Review Assessment: Thoroughness In Paper Reading:**

I read the paper at least twice and used my best judgement in assessing the paper.

---

> ### Author Response · Authors · 2019-11-15
> **Response to review #1**
>
> Thank you for the feedback. Here is a response to your points:
>
> 1. Section 2 helps to understand the intuition for the paper by defining SGD and Adam in a consistent way.
>
> 2. Thanks for the feedback. We have added some clarifications in the paper. z is the input to the learning function. n_t and n_\infty are now defined again in section 3. We removed the equations adding weight decay and nesterov momentum as these were unnecessarily confusing.
>
> 3. The optimal learning rate (eq 1) must take into account both min and max eigenvalues or else one will dominate the time to convergence.  The modification to Adam allows the algorithm to significantly outperform Adam and SGD in computer vision tasks.
>
> 4. The colors were not switched, but the legend order was not consistent. This has been fixed, and the confidence range has been added to the results where available.

---

### Official Review · AnonReviewer2 · 2019-10-22
**Official Blind Review #2**

**Rating:** 1

**Review:**

Summary:

This work proposed a new algorithm called softAdam to unify the advantages of both Adam and SGD. The authors also showed experiments to backup their theoretical results.

Pros:

The authors provided analysis of different algorithms including Adam and SGD on simple quadratic function, then proposed a new algorithm called softAdam which outperforms both Adam and SGD. Experiment results backup their theory.

Cons:

- The novelty of this work is limited. The main contribution of this work is to provide a new adaptive gradient method called softAdam, which changes the update rules for some parameters including \beta. However, neither intuition or theoretical guarantees are provided in this paper. I recommend the authors to add some explanation about why softAdam outperforms other existing algorithms. Besides, the difference between softAdam and original Adam method is little.
- The theoretical analysis about existing adaptive methods provides nothing new. The authors showed some analysis on quadratic model, which is a very simple model and hence can not reflect the true model we face in the practice. I suggest the authors provide some analysis on more general model, including convex functions and non-convex functions.
- The settings of experiments are limited. The authors should at least compare softAdam with other baseline algorithms on some modern deep learning tasks including Imagenet.

Minor comments:

- Page 4, section 3, 'this understanding of has'... lacks object.
- This paper lacks some references in this area.

J. Chen and Q. Gu. Closing the generalization gap of adaptive gradient methods in training
deep neural networks. arXiv preprint arXiv:1806.06763, 2018.
Ward, R., Wu, X. and Bottou, L. (2018). Adagrad stepsizes: Sharp convergence over nonconvex
landscapes, from any initialization. arXiv preprint arXiv:1806.01811 .
Li, X. and Orabona, F. (2018). On the convergence of stochastic gradient descent with adaptive
stepsizes. arXiv preprint arXiv:1805.08114 .



**Experience Assessment:**

I have published one or two papers in this area.

**Review Assessment: Checking Correctness Of Derivations And Theory:**

I carefully checked the derivations and theory.

**Review Assessment: Checking Correctness Of Experiments:**

I assessed the sensibility of the experiments.

**Review Assessment: Thoroughness In Paper Reading:**

I read the paper thoroughly.

---

> ### Author Response · Authors · 2019-11-15
> **Response to review #2**
>
> Thank you for your feedback. Here is a response to your points:
>
> - The novelty of this work is limited....
>
> Thank you for the feedback. We intend future studies that will provide more explanation of why softAdam outperforms other existing algorithms. This is not a trivial problem to answer, as we do not know why SGD and Adam generalize well in the first place. Although theoretical guarantees are not proven in this paper, it can be seen by inspection of the algorithm that the step size is always smaller than the SGD step size, and so the SGD convergence guarantees will be held.
>
> - The theoretical analysis about existing adaptive methods provides nothing new....
>
> The theoretical analysis of the quadratic model here provides intuition for the algorithm’s behavior on convex optimization. An additional study of adaptive methods for non-convex optimization is well described in the reference you provided, “On the convergence of stochastic gradient descent with adaptive stepsizes.”
>
> - The settings of experiments are limited....
>
> CIFAR has provided an adequate testing ground for optimization algorithms for many years. We wanted to have the most comparable results with those other optimization papers.
>
> - This paper lacks some references in this area.
>
> Thank you for these references, we have included them.

---

### Official Review · AnonReviewer3 · 2019-10-23
**Official Blind Review #3**

**Rating:** 3

**Review:**

This paper proposes a new algorithm which brings closer SGD and Adam while also incorporating new corrections to improve behavior of the optimizer in contexts where there is very small or very large eigen values.

Decision

I vote for weak rejection because the core modification proposed to Adam is minor and is mostly supported by intuition and preliminary experiments.

Justification

There is many modifications proposed, but most are secondary corrections for stability, such as the warm-up schedule with the redefinition of beta_2. These modifications could as well be incorporated in Adam without the core modification that is the smoothing presented in section 3. These additional modifications also make it difficult to measure the importance of the core contribution. Without getting rid of them, an ablation study on toy problems (even synthetic data) would be necessary for a better understanding.

In section 3.1, the temporal definition of beta_2t is integrating a warm-up. While the reason for doing so is supported in introduction of section 3, the effect of this modification should be weighted against no warm-up, and also compared with its effect on Adam.

There is an error in algorithm 1. The last element of the last line (Perform the update) should be \alpha (1 - \eta) m_t/d_t. The code in Appendix corroborates this correction. Minor related note, the use of d_t to define the denominator of what d_i represents in section 3 is very confusing. I would suggest to use the ratio notation of d_i from the equations in the algorithm for coherence.

If we get pass the warm-up scheduling, by massaging the equation we get that the algorithm is different from Adam on 2 points, 1) the bias are not corrected and 2) the denominator sqrt(v) + epsilon is replaced by sqrt(v) + sqrt(mean(v) + epsilon^2). I have difficulty convincing myself that smoothing by the average is solving the issues raised in the paper and there is no experiments to study its effect directly.

The experiments are on 3 datasets, but only the computer vision ones are run on multiple architectures. Caption of figure 1 explains that each model is trained 3 times, but the source of variation between each run is not described. Are the models initialized differently? In any case, there is an overlap for 3 of the 5 models between SGD and SoftAdam which makes the comparison rather unconvincing. There is no standard deviation for Adam, and none on Penn Treebank dataset and IWSLT. For a better comparison, all hyper-parameters of the algorithms should be optimized for each run. I understand that SoftAdam is meant to be close to both SGD and Adam, but using the same hyper-parameters may induces misleading results by favoring some (model, optimizer) combination nevertheless.

Minor comments

In section 2, second paragraph, the term 'mini-batch' should be used instead of 'batch'.
In section 2, last sentence, the betas should have no t.
In section 2.1, fourth equation (unnumbered), the eigen vector xi_i is presented as a vector and then used as a scalar. Notation should be uniformed.
In Section 2.1 around equation (2), the use of i and j is incoherent.
In Section 3:
- Overall, this understanding *of* has
- we consider the *an* update

**Experience Assessment:**

I have published one or two papers in this area.

**Review Assessment: Checking Correctness Of Derivations And Theory:**

I assessed the sensibility of the derivations and theory.

**Review Assessment: Checking Correctness Of Experiments:**

I assessed the sensibility of the experiments.

**Review Assessment: Thoroughness In Paper Reading:**

I read the paper thoroughly.

---

> ### Author Response · Authors · 2019-11-15
> **Response to review #3**
>
> Thank you very much for the detailed feedback. Overall, we want to note that the algorithm significantly outperforms Adam and even outperforms SGD in computer vision tasks. The changes here have a significant effect on the generalization performance and constitute novel research.
>
> Below are responses to your points:
>
> - "There is many modifications proposed, but most are secondary corrections for stability, such as the warm-up schedule with the redefinition of beta_2. These modifications could as well be incorporated in Adam without the core modification that is the smoothing presented in section 3. These additional modifications also make it difficult to measure the importance of the core contribution. Without getting rid of them, an ablation study on toy problems (even synthetic data) would be necessary for a better understanding."
> - "In section 3.1, the temporal definition of beta_2t is integrating a warm-up. While the reason for doing so is supported in introduction of section 3, the effect of this modification should be weighted against no warm-up, and also compared with its effect on Adam."
>
> The warmup schedule itself has been studied fairly extensively, especially recently by Liu (2019) and Ma (2019). The modification to the beta_2 for debiasing does not significantly impact the performance and is only used to make the algorithm better align with the theoretical warmup schedule—using the traditional Adam m_t debiasing does not have any impact on the results. We added a comment to this effect.
>
> We also updated the results to specifically use AdamW, which is more comparable to our algorithm.
>
> - "There is an error in algorithm 1. The last element of the last line (Perform the update) should be \alpha (1 - \eta) m_t/d_t. The code in Appendix corroborates this correction. Minor related note, the use of d_t to define the denominator of what d_i represents in section 3 is very confusing. I would suggest to use the ratio notation of d_i from the equations in the algorithm for coherence."
>
> This is correct, thank you for the suggestion. We have adjusted the notation based on your feedback.
>
> - "If we get pass the warm-up scheduling, by massaging the equation we get that the algorithm is different from Adam on 2 points, 1) the bias are not corrected and 2) the denominator sqrt(v) + epsilon is replaced by sqrt(v) + sqrt(mean(v) + epsilon^2). I have difficulty convincing myself that smoothing by the average is solving the issues raised in the paper and there is no experiments to study its effect directly."
>
> (1) the second order bias is corrected, just in a different way. As mentioned before, there is no practical difference between the debiasing methods.
>
> (2) In addition to changing the denominator to sqrt(v) + eta * sqrt(mean(v) + epsilon^2), there is also a scaling of the learning by 1 + eta * sqrt(mean(v)), which changes the implicit learning schedule created by Adam to be like SGD. This is what allows the Adam-like limit of eta→ infinity and the SGDM limit of eta→0. We intend future studies that will dis-entangle these two effects (learning rate schedule versus smoothing direction). The purpose of this paper is to share the algorithm that allows such a study in the first place, which is unique in this regard.
>
> - "The experiments are on 3 datasets, but only the computer vision ones are run on multiple architectures..."
>
> We regret we cannot train on every architecture, but we focused on representative samples for different problems. We have added a comment that the models have different initializations and the hyper-parameters are tuned separately for each algorithm. More runs have been generated to provide standard deviations for the mentioned parameters.
>
> Minor comments have been addressed in the updated draft. Note both z and \xi_i are vectors, so the notation is already correct.

---

### Public Comment · ~Hao_Jin1 · 2019-10-13
**Several Problems about Notations**

What do you mean by n_t and n_\infty in Section 3?

---

> ### Author Response · Authors · 2019-10-15
> **Definition of n_t and n_\infty**
>
> $n_t$ represents the effective number of elements in the average for $v_t$. Using an exponential weighted average as is used in Adam, for large t, $n_t \approx n_\infty \approx 2/(1-\beta_2)$. However, if $t \ll n_\infty$, $n_t \approx t$.

---

### Decision · Program_Chairs · 2019-12-19

**Decision:**

Reject

**Comment:**

The reviewers all agreed that the proposed modification was minor. I encourage the authors to pursue in this direction, as they mentioned in their rebuttal, before resubmitting to another conference.